# A new perspective of permafrost boundaries in France during the Last Glacial Maximum

Kim H. Stadelmaier[1], Patrick Ludwig[1], Pascal Bertran[2,3], Pierre Antoine[4], Xiaoxu Shi[5], Gerrit Lohmann[5,6], and Joaquim G. Pinto[1]

[1]Institute of Meteorology and Climate Research, Karlsruhe Institute of Technology, Karlsruhe, Germany
[2]Inrap, Bègles, France
[3]PACEA, CNRS-Université de Bordeaux, Pessac, France
[4]Laboratoire de Géographie Physique, CNRS, Meudon Cedex, France
[5]Alfred Wegener Institute, Helmholtz Centre for Polar and Marine Research, Bremerhaven, Germany
[6]MARUM and Institute of Environmental Physics, University of Bremen, Bremen, Germany

**Correspondence:** Kim H. Stadelmaier (kim.stadelmaier@kit.edu)

**Abstract.** During the Last Glacial Maximum (LGM), a very cold and dry period around 26.5 to 19 thousand years ago, permafrost was widespread across Europe. In this work, we explore the possible benefit of using regional climate model data to improve the permafrost representation in France, we decipher how the atmospheric circulation affect the permafrost boundaries in the models and test the role of ground thermal contraction cracking in wedge development during the LGM. With these aims, criteria for possible thermal contraction cracking of the ground are applied to climate model data for the first time. Our results show that the permafrost extent and ground cracking regions deviate from proxy evidence when the simulated large-scale circulation in both global and regional climate models favours prevailing westerly winds. A colder and with regard to proxy data more realistic version of the LGM climate is achieved given more frequent easterly winds conditions. Given the appropriate forcing, an added value of the regional climate model simulation can be achieved in representing permafrost and ground thermal contraction cracking. Furthermore, the model data provide evidence that thermal contraction cracking occurred in Europe during the LGM in a wide latitudinal band south of the probable permafrost border, in agreement with field data analysis. This enables the reconsideration of the role of sand wedge casts to identify past permafrost regions.

## 1 Introduction

Permafrost is an important component of the climate system and is particularly sensitive to variations of climate. Permafrost is defined as ground - including soil, rock, ice and organic material - that remains at or below $0\,°C$ for at least two consecutive years (e.g., van Everdingen, 2005). In recent decades, thawing of permafrost soils has affected many high-latitude regions, and thawing is most likely to accelerate in the near and long-term future (IPCC, 2013, 2019; Harris et al., 2009). While enhanced greenhouse gas forcing leads to warming temperatures and thus to permafrost thawing, the thawing itself leads to the release of greenhouse gases that were previously bound within the frozen soils. Thereby, the greenhouse effect is enhanced which leads to further warming of the climate, in a positive feedback (e.g., IPCC, 2019; Liu and Jiang, 2016a; Schuur et al., 2015). Current climate model simulations project a large range of uncertainties regarding the decrease of permafrost areas (e.g., IPCC,

2019; Schuur et al., 2015). The models are calibrated for present-day conditions, under which they are well tested. However, the responses of the models to the same forcing vary by several orders of magnitude (e.g., Braconnot et al., 2012; Cleator et al., 2020; IPCC, 2013, 2019). It is therefore necessary to evaluate the climate models under a wider range of climate conditions. This can be achieved by simulating past climates and comparing the results with proxy evidence from Quaternary sequences (Braconnot et al., 2012; Harrison et al., 2015; Smerdon, 2012).

During the Last Glacial Maximum (LGM; Clark et al., 2009; Mix et al., 2001) corresponding to around 26.5 to 19 thousand years (ka) before present, huge ice sheets covered large parts of the Northern Hemisphere, modifying the surface albedo and orography (Hughes et al., 2015; Ullman et al., 2014), and enhanced sea ice cover modified heat fluxes between the ocean and atmosphere (Flückiger et al., 2008). During the coldest phase of the LGM, the sea level was about 130 m lower than today (Lambeck et al., 2014) and the greenhouse gas concentrations were at a historical minimum with values less than half of its current concentrations (Clark et al., 2009; Monnin et al., 2001). Lower greenhouse gas concentrations favoured the growth of C4 over C3 plants (Prentice and Harrison, 2009), although only C3 plants have actually been identified in European loess (Hatté et al., 1998, 2001). Globally, this hampered the development of trees (Woillez et al., 2011), resulting in less productive terrestrial ecosystems and more open vegetation (Bartlein et al., 2011). Ultimately, this resulted in easily erodible soils, whose contribution to the dust cycle increased (Prospero et al., 2002; Ray and Adams, 2001). These boundary conditions and forcing led to a substantially different climate than today. In general, the LGM was a colder, drier and windier period in Earths' history compared to the recent climate (e.g., Annan and Hargreaves, 2013; Bartlein et al., 2011; Löfverström et al., 2014). The global and annual mean surface air temperatures were about 4 °C colder than today, with differences reaching up to 14 °C close to the LGM ice sheets e.g., in Central Europe (e.g., Annan and Hargreaves, 2013; Bartlein et al., 2011; Clark et al., 2009; Pfahl et al., 2015; Ludwig et al., 2017). The atmospheric circulation in the North Atlantic region varied considerably from the current conditions mainly due to the direct influence of the altered topography by ice sheets (Justino and Peltier, 2005; Merz et al., 2015). A planetary large-scale atmospheric wave with an amplitude much larger than today was induced, with a deep trough downstream of the Laurentide ice sheet. This led to a generally more zonal orientation of the North Atlantic jet stream (Löfverström et al., 2014). Additionally, the jet was enhanced and its position was shifted southward (e.g., Li and Battisti, 2008; Merz et al., 2015; Pausata et al., 2011). The storm track during the LGM evolved accordingly (e.g., Löfverström et al., 2014; Ludwig et al., 2016; Raible et al., 2020), and extreme cyclones were more intense and characterized by less precipitation (Pinto and Ludwig, 2020). Thus, cyclones were able to trigger more frequent dust storms during the LGM (Antoine et al., 2009; Pinto and Ludwig, 2020; Sima et al., 2009). Besides these dust storms, easterly winds induced by an anticyclone over the Fennoscandian ice sheet (FIS) were another important factor for the deposition of loess in Central and Western Europe (Krauß et al., 2016; Schaffernicht et al., 2020; Stevens et al., 2020) as well as westerly to northwesterly winds (e.g., Renssen et al., 2007; Schwan, 1986, 1988).

At the same time, adjacent areas south of the FIS were widely affected by permafrost (Kitover et al., 2013; Levavasseur et al., 2011; Saito et al., 2013; Vandenberghe et al., 2014; Washburn, 1979). The past permafrost distribution is usually inferred from the occurrence of a variety of fossil periglacial features, among which ice wedge pseudomorphs are the most reliable and widespread (e.g., Bertran et al., 2014; Huijzer and Isarin, 1997; Péwé, 1966; Vandenberghe, 1983; Vandenberghe et al., 2014).

Ice wedges develop within perennially frozen ground, when temperature drops quickly and the ground experiences thermal contraction cracking. Annual frost cracks that reach downward into the permafrost are a few mm wide. They get filled with snowmelt water that freezes into ice veins. Repeated cracking over years at the same location add ice veins that constitute ice wedges (e.g., Harry and Gozdzik, 1988; Murton, 2013). Ice wedge pseudomorphs observed from the LGM in Europe were formed when the ice melted and the cavities were filled by collapsing soil materials. Today, ice wedges are mostly active in continuous permafrost environments (Fortier and Allard, 2005; Kokelj et al., 2014; Matsuoka et al., 2018; Péwé, 1966). Open cracks may also be filled with wind-blown sand, which gives rise to sand wedges, or by both ice and sand (composite wedges). Active sand wedges are found today primarily in areas characterised by continuous permafrost and limited snow and vegetation cover (i.e. the polar deserts), and with local sources of aeolian sediments such as in Antarctica (Bockheim et al., 2009; Levy et al., 2008; Murton et al., 2000; Péwé, 1959). Ground cracking is often restricted to the active layer (i.e. the surface layer subjected to seasonal freezing and thawing) in the areas underlain by "warm" permafrost (i.e. at a temperature close to 0°C) and south of the permafrost border. Thin cracks develop and are referred to as seasonal frost cracks. However, Wolfe et al. (2018) showed that large shallow sand wedges can also develop in Canada in areas with deep seasonal ground freezing (i.e. without perennially frozen ground) in mineral soils close to dune fields, which provide abundant sand to fill the cracks.

Thermal contraction cracking of the ground is the causal factor that leads to ice (or sand) wedge growth. Ecological factors such as type of vegetation cover and thick snow cover often limit thermal contraction cracking as they may prevent the cooling of the ground. This is the case in current densely vegetated areas that insulate the ground and trap snow (e.g. shrub tundra and taiga; Kokelj et al., 2014; Mackay and Burn, 2002). Conversely, cracking can occur at low frequency in mid-latitude, cool temperate regions in grounds devoid of tall vegetation and snow, particularly in roads and airport runways (Barosh, 2000; Okkonen et al., 2020; Washburn, 1963).

Many attempts at reconstructing the past permafrost distribution in Europe using field proxies have been performed during the last decades. Based on the assumption that both active ice wedges and sand wedges are associated with continuous permafrost and possibly with widespread discontinuous permafrost (Burn, 1990; Romanovskij, 1973), some of the earliest reconstitutions as reported by Vandenberghe et al. (2014) proposed that Europe was affected by permafrost as far south as the 43.5° N latitude. However, a detailed analysis of periglacial features in France by Andrieux et al. (2016b, 2018) demonstrated that typical ice wedge pseudomorphs are exclusively found north of latitude 47.5° N, whereas sand wedge casts occur at lower latitude at the periphery of aeolian sandsheets. A correlation between wedge depth and latitude has also been highlighted, which strongly suggests that the southernmost shallow sand wedges developed in regions where perennial ice could not form, i.e. without permafrost or with sporadic permafrost. A similar pattern has also been highlighted in China by Vandenberghe et al. (2019). The sand wedges reach up to 1m wide in southwest France near 45°N in the periphery of coversands. Optically stimulated luminescence dating of the sand fill by Andrieux et al. (2018) have demonstrated that these large epigenetic sand wedges resulted from repeated periods of growth throughout the Last Glacial.

Multiple attempts have also been performed to infer the LGM permafrost occurrence from climate model data. Liu and Jiang (2016b) considered both direct and indirect methods. The simplest indirect method is based on the modelled mean annual air temperature (MAAT). Threshold values for permafrost occurrence were adapted according to ground texture (Vandenberghe

et al., 2012). However, this method only provides a rough estimate of permafrost extension, since a variety of other factors are known to impact ground temperatures, including water content, vegetation and snow cover. Particularly, the insulating effects of snow and vegetation cover may be responsible for an offset of up to $6\,°C$ between MAAT and the mean annual ground surface temperature (MAGST). On the other hand, variations in ground thermal conductivity (depending on texture and water content) may result in an offset of $2\,°C$ between MAGST and the temperature at the top of permafrost (TTOP) (e.g., Smith and Riseborough, 2002; Throop et al., 2012). A refined indirect method to derive permafrost occurrence from climate model data is the use of the surface frost index (SFI, Nelson and Outcalt, 1987), which corresponds to the ratio between frost and thaw penetration depths and takes the effects of snow in account. The SFI has been used in several studies, with only minor changes of the original method. For example, monthly model output was used instead of summing up daily air temperatures (e.g., Frauenfeld et al., 2007; Liu and Jiang, 2016b). Slater and Lawrence (2013) weighted the snow depth for each month to consider snow accumulation effects, while Stendel and Christensen (2002) replaced the surface air temperature with the temperature of their deepest simulated ground layer (5.7 m deep) to investigate permafrost degradation due to current global warming. They pointed out the advantage of taking simulated ground temperatures, where insulation effects of snow and vegetation cover are explicitly taken into account by the models, render empirical approaches redundant. For the direct method, the modelled ground temperatures below $0\,°C$ are used to diagnose permafrost. The studies differ slightly in the depth of the considered ground temperatures (e.g., Liu and Jiang, 2016a, b; Saito et al., 2013; Slater and Lawrence, 2013).

Studies investigating the permafrost limits during the LGM using global climate simulations have failed so far to reproduce appropriately the permafrost extent as reconstructed from field proxies (e.g., Andrieux et al., 2016b; Levavasseur et al., 2011; Ludwig et al., 2017). However, there is evidence for improvements when using the data from regional climate simulations (e.g., Ludwig et al., 2017, 2019). The aim of this study is (1) to explore the possible benefit of using regional climate model data to improve the permafrost representation over France, (2) to decipher how the atmospheric circulation affect the permafrost boundaries in the models and finally, (3) to test the role of ground thermal contraction cracking in wedge development during the LGM.

In section 2 we introduce the adaptions made to the regional climate model to be compliant with LGM boundary conditions and describe the global simulations that provide the initial and boundary conditions. Further, we give an overview of the different methods used to derive the LGM permafrost distribution in France. In section 3 we describe the general characteristics and differences of the LGM climate based on the global and regional climate model data and present the permafrost and ground cracking distribution based on regional climate model data. Finally, we discuss and summarize the results in section 4.

## 2 Data and Methods

In this study, LGM simulations of two global climate models, namely MPI-ESM-P (MPI; Jungclaus et al., 2013; Stevens et al., 2013) and ECHAM6-FESOM (AWI; Sidorenko et al., 2015; Lohmann et al., 2020), are dynamically downscaled with the Weather Research and Forecast model (WRF; Skamarock et al., 2008). Both global models share the same atmospheric component ECHAM6 but different modules for the ocean. The MPIOM (Marsland et al., 2003) is coupled within the MPI-

ESM-P, forming the well established global climate model that took part in several CMIPs. In AWI-ESM, the FESOM ocean model(Wang et al., 2014) featuring an unstructured mesh as well as a multi-resolution approach is used with a relatively high resolution of less than 30 km north of 50° N. The atmospheric grid applied in the MPI and AWI experiment is T63 (roughly 1.9 degree spatially) with 47 unevenly distributed vertical levels. The simulations follow either the PMIP3 (MPI; Braconnot et al., 2012; http://pmip3.lsce.ipsl.fr/) or PMIP4 protocol (AWI; Kageyama et al., 2017), where the boundary conditions (solar constant, orbital parameters, greenhouse gases) are set accordingly to best estimate the LGM boundary conditions. The AWI-ESM has been used in the recent CMIP6/PMIP4 intercomparisons (Brierley et al., 2020; Keeble et al., 2021; Kageyama et al., 2021) and was applied for the LGM (Lohmann et al., 2020). The ice sheet provided for PMIP3/CMIP5 LGM experiments is a blended product obtained by averaging three different ice sheet reconstructions: ICE-6G v2.0 (Peltier et al., 2015), MOCA (Tarasov and Peltier, 2003) and ANU (Lambeck et al., 2002), whereas the LGM topography in AWI experiment is configured based on the ICE6G reconstruction (Peltier et al., 2015). For the recent climate, the pre-industrial period (PI), corresponding roughly to 1850, is used as a reference. The simulations follow again the PMIP3 (MPI; Taylor et al., 2012) or PMIP4 protocol (AWI; Eyring et al., 2016).

To account for model uncertainties, outputs from these global LGM simulations are used to drive the regional WRF simulations. The atmospheric boundary conditions are updated every six hours, SST and sea ice cover are updated daily. Apart from the different forcing, the set up of the two regional simulations is identical. The coastlines, ice sheet extent, trace gas conditions and orbital parameters are adapted to LGM values according to the PMIP3 protocol (Ludwig et al., 2017). Modifications to the Alpine ice sheet are implemented according to Seguinot et al. (2018). Land use and vegetation cover is taken from the CLIMAP data set (CLIMAP Project Members, 1984). An overview of the used parametrization schemes in the WRF simulations is given in Table 1. Most important for the representation of the ground characteristics is the parametrization of the land surface, for which we used the unified Noah Land Surface Model (Tewari et al., 2004). Based on 19 different soil types, various ground parameters (e.g. ground thermal conductivity) are set and used for the calculations of ground temperatures and moisture for each grid point. More details can for example be found in Chen and Dudhia (2001) and Niu et al. (2011) and references therein. The first model domain covers large parts of Europe with a horizontal resolution of 50 km (see Fig. 1) and 35 vertical layers up to 150 hPa. The integration time step is 240 seconds. The second, nested domain covers southern parts of the FIS, the Alps and France, where the latter represents the target region to assess the LGM permafrost limits in this study. Here, the horizontal resolution is 12.5 km and the integration time step is 48 seconds. The soil is separated into four layers, with representative depths of 5, 25, 70, and 150 cm. A total of 32 years are simulated for each global forcing simulation. The first two years are used as a spin up phase and are excluded from further analysis. This way it is ensured, that the atmosphere and soil properties and processes are in equilibrium.

The permafrost distribution is derived from climate model data using the three different methods described in Sec. 1. For MAAT, the 2 m air temperature is considered. Threshold values were derived from data compiled from studies in current Arctic regions, where continuous permafrost is inferred for MAATs $< -8 \pm 2\,°C$, whereas discontinuous permafrost requires MAATs $< -4 \pm 2\,°C$ (e.g., Smith and Riseborough, 2002; Vandenberghe et al., 2012). The surface frost index (SFI) is based on the annual freezing and thawing degree-days (DDF and DDT, respectively) being the sum of daily air temperatures below

or above 0 °C, respectively:

$$\text{SFI} = \sqrt{\text{DDF}}/(\sqrt{\text{DDF}} + \sqrt{\text{DDT}}).$$

An SFI between 0.5 and 0.6 indicates sporadic permafrost, between 0.6 and 0.67 discontinuous permafrost, and above 0.67 continuous permafrost (e.g., Nelson and Outcalt, 1987; Stendel and Christensen, 2002). For this indirect method, we use ground temperatures of the third layer at 78 cm and 70 cm for the global and regional simulations, respectively. With the direct method, permafrost is inferred when ground temperatures are at or below 0 °C.

Beyond the permafrost indices, ground cracking is assumed to be possible when two conditions derived from fieldwork by Matsuoka et al. (2018) are fulfilled simultaneously: A daily mean soil temperature below -5 °C at a depth of 1 m and a temperature gradient in the upper meter of the ground below -7 °C m$^{-1}$. These minimum values might represent shallow cracking within the active layer/seasonally frozen layer and can be compared against the sand wedge distribution. Conditions for intensive and deep thermal contraction cracking ($T_{100} = -10$ °C and $G_{AL} = -10$ °Cm$^{-1}$) are tested in regard to the ice-wedge pseudomorph distribution in France. Due to higher ice content and higher organic carbon content of the ground, these values do not necessarily correspond exactly to those of France during the Pleistocene. We use the third soil layer again, with depths of 78 cm in the global simulations and of 70 cm in the regional simulations.

To evaluate the model simulations, the distribution of ice wedge pseudomorphs and sand wedges after Andrieux et al. (2016b) and Isarin et al. (1998) are considered.

## 3 Results

### 3.1 Global Boundary Conditions

In this section, we present the large-scale characteristics of the LGM climate derived from global climate model data that is used for dynamical downscaling in comparison with the respective PI simulations. It is important to investigate the climatic mean state and possible biases of the global projections in order to be able to interpret the regional simulations accurately.

Both global models simulate colder annual mean SSTs under LGM than under PI conditions (c.f. Fig. 2 a and b). For the MPI model, a limited area with enhanced SSTs is simulated over the North Atlantic. This does not match with proxy data (MARGO Project Members, 2009) and is a known issue for this and other PMIP3 models (e.g., Wang et al., 2013; Ludwig et al., 2016, 2017). The AWI simulation does not show this warm anomaly over the North Atlantic and the SSTs are generally colder. In the Arctic Ocean, the SSTs in the AWI simulation are considerably higher than in the MPI simulation. This can be explained by the sea ice cover, which is lower in the AWI LGM simulation.

The analysis of wind speed at 300 hPa gives insights into the jet stream structure and strength, which are dominant factors of the atmospheric large-scale circulation over the North Atlantic / European region. In agreement with Li and Battisti (2008), both models show a stronger jet under LGM conditions compared to the simulations under PI conditions (c.f. Fig. 2 c and d). This is particularly the case over the North Atlantic, south-eastward of the Laurentide ice sheet, where the wind speed is up to 14 m/s higher in annual mean for LGM. On the other hand, the wind speed on both the southern and northern flanks of

the jet stream is actually 2-4 m/s weaker during the LGM, indicating a more constrained large-scale flow. Even though the wind anomalies are quite similar for both GCMs, the actual structure is dissimilar: While for the MPI simulations the jet is less constrained and deflected to the north, reaching Europe at the latitude of Ireland, the jet stream in the AWI GCM reaches Europe at the latitude of the Iberian Peninsula and France and extends farther into the continent. In general, the simulated winds speeds at 300 hPa in the AWI model are weaker compared to the MPI model (not shown). The zonal structure of the wind speed anomalies identified for the AWI simulations is more similar to the ensemble mean of CMIP5 models (e.g., Ludwig et al., 2016) than the MPI anomaly pattern.

## 3.2 Climate of the Regional Simulations

Based on the GCM simulations, we obtain two different variants of the regional LGM climate in western Europe. The results are shown primarily for the larger domain of the regional simulations, as the climate of the high resolution simulations yields a similar structure.

The annual mean 2 m air temperature is considerably lower in the WRF-MPI than in the WRF-AWI simulation (c.f. Fig. 3). The biggest differences are identified near the ice sheet margin, with almost 10 °C in annual mean. The sign and pattern of the differences are visible in all seasons, but winter air temperatures clearly diverge most. For the summer, air temperatures in both models are more similar to each other. These differences can be partly attributed to the snow cover: Except for summer, almost the entire region is covered by snow in WRF-MPI, even though a snow height of several metres is only reached over the FIS and the Alpine region (c.f. Fig. S3). WRF-AWI shows markedly higher snow accumulation over the ice sheets with differences of more than 20 m compared to the WRF-MPI simulation, but generally less snow cover in southern and Central Europe. Differences amount to 20 % less snow cover in WRF-AWI in annual mean and to 40 % in both spring and winter. In summer, only the ice sheets are snow covered in both simulations, and the differences are thus negligible. Nevertheless, more precipitation is simulated over Europe in the WRF-AWI simulation (c.f. Fig. 4). High precipitation amounts are either orographically induced as for precipitation over the Alps and over the FIS or they are associated with the moisture availability of the North Atlantic.

The absolute annual mean wind field and the associated differences are depicted in Fig. 5. Both simulations show strongest winds south of the FIS in annual mean and winter, but with a notably enhanced pattern in WRF-MPI, where this also holds for each season. These winds are easterlies/northeasterlies. In contrast, westerly winds from the North Atlantic are stronger in WRF-AWI, thus transporting heat and moisture towards Europe. During winter, the westerly winds are directed towards the centre of the domain in WRF-AWI, whereas in WRF-MPI the winds have a more southwestern component, directed towards the outside of the domain. In summer, both WRF-MPI and WRF-AWI simulations are charaterized by westerly winds from the North Atlantic. Again, winds from the FIS are blowing south- and southeastwards, but the summer wind speeds are consistently weaker than in winter for both simulations. These wind fields are induced by the large-scale circulation in the global forcing simulations. In fact, the northerly and easterly components are predominant in the MPI simulation (Ludwig et al., 2016), whereas southerly and westerly components occur more often in the AWI simulation. This is in accordance

with the jet structure in both global simulations. As the influence of the ice sheet is higher in the (global and regional) MPI simulations, this is consistent with a partially drier and generally colder climate in western Europe during LGM.

### 3.3 Permafrost and Ground Cracking Distribution

The permafrost distribution of the global and regional simulations based on the SFI is depicted in Fig. 6. The permafrost extent based on the AWI-ESM and WRF-AWI simulations does not reach farther south than the ice sheet, apart from the Alps in WRF-AWI. A modest increase in the permafrost area is simulated by the global MPI simulation. Here continuous permafrost is still limited to the ice sheet, but sporadic permafrost is slightly more widespread. The WRF-MPI simulation shows a larger permafrost extent. In eastern Europe, the distribution of ice wedge pseudomorphs (Isarin et al., 1998) strictly overlaps that of modelled continuous permafrost in the selected layer with a depth of 70 cm. In western Europe, field evidence for permafrost exceeds the modeled sporadic permafrost to the south. The conditions for discontinuous and sporadic permafrost are rarely fulfilled in all simulations.

The results of the direct method (c.f. Fig. S4) using long-term mean annual soil temperatures agree with the permafrost extent based on the SFI. However, the different types of permafrost cannot be distinguished by this method, leading to a permafrost line that corresponds to that of the sporadic permafrost based on the SFI.

Permafrost estimations based on MAAT are limited to the permanent ice areas during the LGM in all four simulations (c.f. Fig. S5). Despite of the different regional climate, the hereby reconstructed permafrost boundaries closely resemble each other. The regional climate model simulations show some additional permafrost areas, which are related to higher orography, especially at the Alps, and, in WRF-MPI, also at the Pyrenees and the Massif Central (c.f. Fig. 1 and Fig. S2). These mountainous areas are not adequately resolved in the global forcing simulations because of the coarse horizontal grid spacing.

Conditions for thermal contraction cracking after Matsuoka et al. (2018) have been tested based on the global and regional climate model data. Examples of how the soil temperature and the gradient develop over two consecutive years in France (locations A and B of Fig. 1) are shown in Fig. 7. Time series of the entire simulation periods for these locations can be found in Fig. S6. The two minimum criteria (a) ground temperature at -1 m below -5 °C and (b) temperature gradient in the upper meter of ground greater than -7°C m$^{-1}$) are fulfilled when both curves reach below the depicted reference line. In the WRF-MPI simulation (Fig. 7 a and c), this is the case several times in both years and locations, but not in the WRF-AWI simulation.

For each grid cell, the number of days per year when the thermal contraction cracking criteria (Matsuoka et al., 2018) are fulfilled, is translated into heat maps for each simulation. The results of the minimum conditions for (shallow) cracking are shown in Fig. 8, those of the conditions for intensive and deep cracking in Fig. S7. Whereas the permafrost area is much smaller in the global models than their respective regional counterpart, the opposite is the case for thermal contraction cracking areas. The global AWI simulation almost meets the boundaries of sand wedge occurrence. According to the global MPI simulation, thermal contraction cracking would have been possible as far south as the Iberian Peninsula, where no field evidence for it has been found so far. This can be associated with the lower resolution of the global simulations. Here, the Pyrenees are not resolved adequately in the model and do not act as a natural barrier for cold air arriving from the North, which can thus reach further south in the GCMs (c.f. Fig. 1 and Fig. S2). As for the permafrost distribution, also the possible thermal contraction

cracking occurrence is poorly represented in WRF-AWI and is not able to explain the occurrence of wedges in mid and southern France. By contrast, the WRF-MPI simulation agrees well with proxy evidence. Apart from two sand wedges in the lower Rhône valley, the conditions for thermal contraction cracking are found in the simulation in the area where the features are found. This spatial coherence is further improved in the high-resolution simulation (c.f. Fig. 8 e), which can be primarily attributed to a higher resolved orography (c.f. Fig. 1 b). Also the conditions for deep ground cracking are represented best in the regional WRF-MPI simulation. The heat maps show that those conditions did not occur in southwestern France, in agreement with the field data. In this area, the sand wedge casts do not exceed a depth of two metres and ice wedges pseudomorphs are not mapped at all (Andrieux et al., 2016b).

## 4    Summary and Discussion

In this study we explore the benefit of using regional climate model data for the delimitation of the LGM permafrost distribution in comparison with field proxies in France. The main findings can be summarized as follows.

(1) The SFI is suitable to infer LGM permafrost from climate model data. The results based on the SFI are supported by the direct method as the boundaries between permafrost occurrence and absence as indicated by the SFI fully match the permafrost border derived from the annual mean ground temperature. Among the models used, the SFI based permafrost extent of the regional WRF-MPI simulation best agrees with proxy data and is clearly improved over its global counterpart.

(2) The thermal contraction cracking may have occurred much further south than the simulated permafrost limits, in a context of low and sparse vegetation. The southern extent of sand wedges and that of ice-wedge pseudomorphs in France as delineated by Andrieux et al. (2016b) fit well with the boundaries of LGM thermal contraction cracking derived from the regional WRF-MPI simulation based on the criteria for shallow and for deep cracking after Matsuoka et al. (2018), respectively. In contrast, the global MPI simulation does not resolve orographic features (e.g. the Pyrenees and the Rhone valley) sufficiently, leading to a possible southward airflow transporting cold air across France to Spain, and allows ground cracking to occur at excessively low latitudes.

(3) The obtained estimates for the possible location of permafrost is consistent with the hypothesis proposed by Andrieux et al. (2016b, 2018), who suggest that sand wedges did not exclusively form in permafrost areas during the LGM, but also developed within deep seasonally frozen ground. Contrary to what occurs today in large Arctic areas underlain by permafrost, where ground insulation is limited by dense vegetation (shrub tundra, taiga) and snow cover prevents ground cracking and limits the growth of ice wedges (existing ice wedges that have formed in relation to different climatic or ecological conditions do not melt but are dormant), ice wedge growth in permafrost areas in France during the LGM was rapid because thermal conditions leading to ground cracking occurred with high frequency. Large ice wedges (which after thawing developed into recognizable pseudomorphs) would have formed in permafrost where it was cold enough in winter to crack. Simulations show that periods of winter ground temperatures below -10°C at 1m depth could occur in the discontinuous and sporadic permafrost zone, suggesting that thermal contraction cracks were possibly not restricted to the active layer but could propagate into the permafrost in these areas leading to the development of ice wedges. The regional WRF-MPI simulations best match the proxy-

based permafrost reconstruction. The agreement with the proxies is better in eastern Europe, even though the availability of field data remains scarce in that region compared to western Europe. The presence of ice wedge pseudomorphs in northern France actually shows that permafrost must have extended at least 150 km further south than simulations suggest.

The consideration of two global models enables the quantification of uncertainties associated with the large-scale flow considitions under LGM. The global MPI and AWI simulations differ in their atmospheric flow and jet structure. In AWI, the westerly flow dominates so that moisture and heat is transported from the North Atlantic towards Europe. This large-scale circulation is in good agreement with the multi-model mean of the CMIP5/PMIP3 and CMIP6/PMIP4 models, whereas the MPI simulation exhibits a more northward jet stream and suggests a stronger ice sheet influence through prevailing north- and northeasterly

winds (Kageyama et al., 2021; Ludwig et al., 2016). Considering that the regional WRF-MPI simulation is largely in agreement with proxy evidence for both the permafrost and ground cracking extent, we assume that the large-scale circulation of the LGM is reflected more accurately in this simulation. For wind and air pressure so far only indirect proxy evidences exist, e.g., the reconstruction of easterly wind directions from sediments across the European loess belt (Dietrich and Seelos, 2010; Krauß et al., 2016; Römer et al., 2016). Because of the drier conditions with less vegetation and higher wind speeds, dust events

occurred frequently during the LGM. This is reflected by the thick loess deposits in western and Central Europe, which form the European less belt (e.g., Lehmkuhl et al., 2016). Recent studies similarly support the hypothesis that, besides individual cyclone events, easterly winds induced by a semi-permanent anticyclone over the FIS were an important component for the glacial dust cycle (e.g., Raible et al., 2020; Schaffernicht et al., 2020; Stevens et al., 2020).

Overall, the new regional climate simulations largely reconcile the field data and enables the reconsideration of the significance

of ice wedge pseudomorphs and sand wedge casts for understanding past climate variations. Field data still suggest a wider extension of permafrost in western Europe than shown by the simulations, however, analysing the southern extent of thermal contraction cracking completes the picture. Various factors may account for a remaining gap between proxy and model data. These factors include:

(1) The ground thermal conductivities used in the models may not be perfectly adequate. For fine-grained soils such as loess

(in which many ice wedge pseudomorphs have been reported), this could lead to a slightly colder ground temperature, although this effect is assumed to have been minor.

(2) Snow depth and snowpack properties (e.g., Royer et al., 2021) are a very sensitive factor for permafrost and some snow processes are not considered in the models. This may explain some of the discrepancies between field data and simulations. Snow sweeping by the wind at some sites, especially on plateaus, may have led to local permafrost development. However, it

should be mentioned that pseudomorphs have been described in the Last Glacial floodplains in the Paris Basin (e.g., Bertran et al., 2018), i.e. in places a priori favourable to snow accumulation.

(3) Data from loess sections in northern France (Antoine et al., 2003, 2014) and Germany (Meszner et al., 2013) show that the main phases of ice wedge development occurred between 30 ka and 24 ka. This period called the Last Permafrost Maximum (LPM, Vandenberghe et al., 2014) covers short and very cold events, which resulted in wider permafrost extension than during

the LGM sensu stricto. However, boundary conditions for the simulations are only known accurately at 21 ka.

To conclude, the combination of the well established permafrost index SFI and the criteria for thermal contraction cracking

by Matsuoka et al. (2018), both based on regional climate model data, provides new possibilities for the estimation of the permafrost extent and the interpretation of ice and sand wedges, especially for paleoclimate applications. In this context, the use of regional climate model simulations with a highly resolved orography is clearly beneficial (e.g., Ludwig et al., 2019) and should be considered for other regions than western Europe.

*Code and data availability.* The source code of AWI-ESM is available from the AWI based svn repository (https://swrepo1.awi.de/projects/awi-cm/). The data are available at Kageyama et al. (2021). Data of the PI and LGM simulations of the MPI-ESM-P are available at Jungclaus et al. (2012a) and Jungclaus et al. (2012b), respectively. The WRF data will be archived at DKRZ (German Climate Computing Centre) and are available from the corresponding author upon request. PMIP3 boundary conditions can be obtained at https://pmip3.lsce.ipsl.fr/ (last access: 20 May 2021). Vegetation cover and land use data from CLIMAP (1984) can be obtained at https://iridl.ldeo.columbia.edu/SOURCES/.CLIMAP/.LGM/ (last access: 20 May 2021). The database of ice wedge pseudomorphs and sand wedge casts in France is available at https://afeqeng.hypotheses.org/487 and discussed in Andrieux et al. (2016a). The database of European ice wedge pseudomorphs and sand wedge casts is available at http://nsidc.org/data/ggd248.html.

*Author contributions.* PL, PB and JGP designed the concept of the study. PL adjusted the WRF model for LGM applications. KHS performed the regional simulations with the WRF model, analysed the data and created the figures. XS and GL provided data of the global AWI simulations. PB provided the proxy data. KHS wrote the first draft of the manuscript. All authors contributed with discussions and revisions to the final manuscript.

*Competing interests.* The authors declare that they have no conflict of interest.

*Acknowledgements.* KHS and JGP thank the AXA Research Fund for support. PL is supported by the Helmholtz Climate Initiative REKLIM (regional climate change; https://www.reklim.de/en). KHS, PL, XS and GL thank the German Climate Computing Centre (DKRZ, Hamburg) for providing computing resources. This study is a contribution to the PALEOLINK project (http://pastglobalchanges.org/science/wg/2k-network/projects/paleolink/intro) within the PAGES 2k Network, as well as a contribution to PalMod and PACMEDY projects funded by the BMBF.

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

**Table 1.** Physical parametrization schemes used for the regional simulations.

|  | WRF-namelist option | Scheme and Reference |
|---|---|---|
| Micro Physics | mp_physics = 4 | WRF Single-moment 5-class Scheme |
|  |  | Hong et al. (2004) |
| Radiation | ra_sw_physics = 4 | rrtmg scheme |
|  | ra_lw_physics = 4 | Iacono et al. (2008) |
| Surface Layer | sf_sfclay_physics = 1 | Revised MM5 Monin-Obukhov scheme |
|  |  | Jiménez et al. (2012) |
| Land Surface | sf_surface_physics = 2 | unified Noah Land Surface Model |
|  |  | Tewari et al. (2004) |
| Planetary Boundary Layer | bl_pbl_physics = 1 | Yonsei University Scheme (YSU scheme) |
|  |  | Hong et al. (2006) |
| Cumulus Parametrization | cu_physics = 1 | Kain-Fritsch Scheme |
|  |  | Kain (2004) |

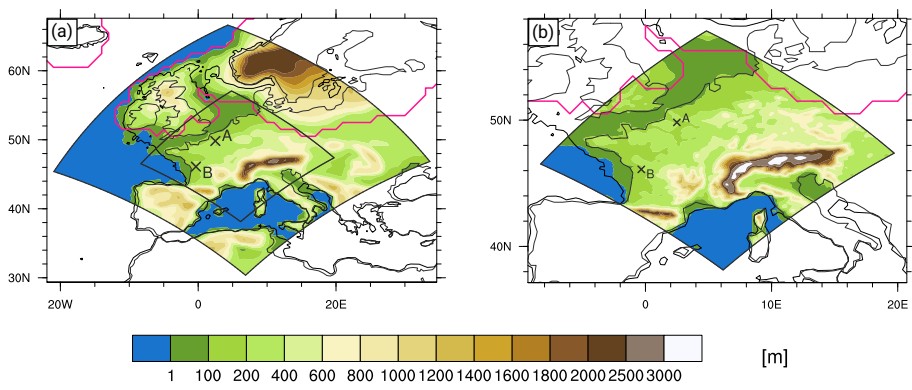

**Figure 1.** WRF model domain orography with LGM coastline (black line) and LGM ice sheet (pink line). *(a)* Domain 1 with 50 km grid spacing, *(b)* Domain 2 with 12.5 km grid spacing. Locations A and B on the map refer to time series of soil temperatures and soil temperature gradients in Fig. 7 and Fig. S6

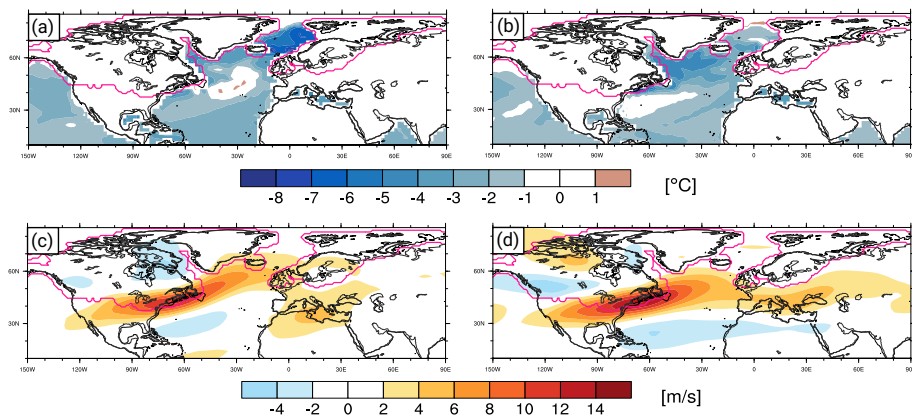

**Figure 2.** *(a)* Distribution of annual mean SST differences between the global MPI simulations under LGM and PI conditions. *(b)* as *(a)* but for the global AWI simulations. *(c)* and *(d)* as *(a)* and *(b)* but for the annual mean wind speed in 300 hPa.

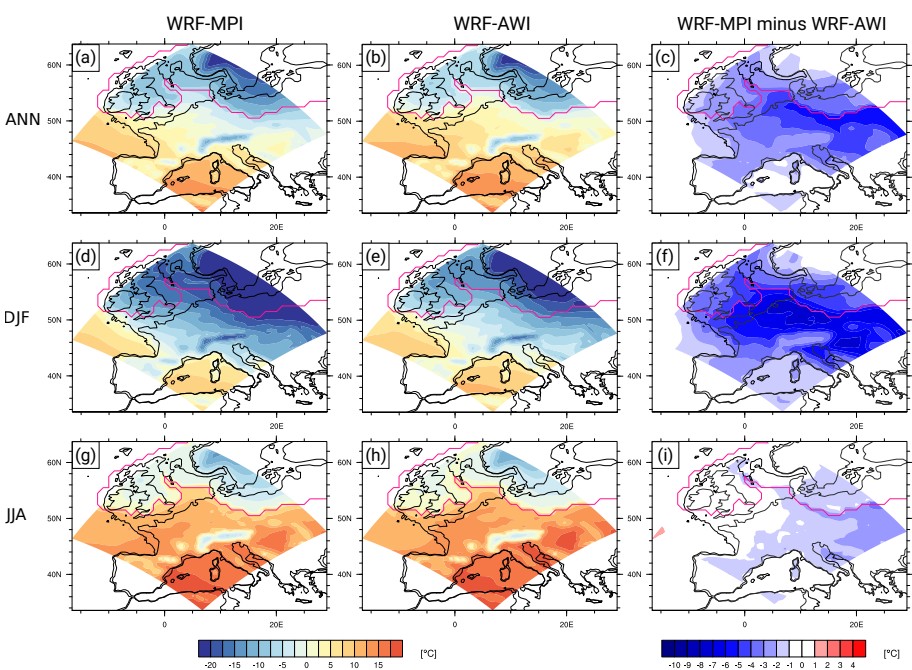

**Figure 3.** Distribution of 2 m air temperature in annual *(a-c)* and seasonal winter *(d-f)* and summer *(g-i)* means as simulated with the regional WRF model with MPI forcing *(a, d and g)* and with AWI forcing *(b, e and h)* and their differences *(c, f and i)*. Black line: LGM coastline, pink line: LGM ice sheet.

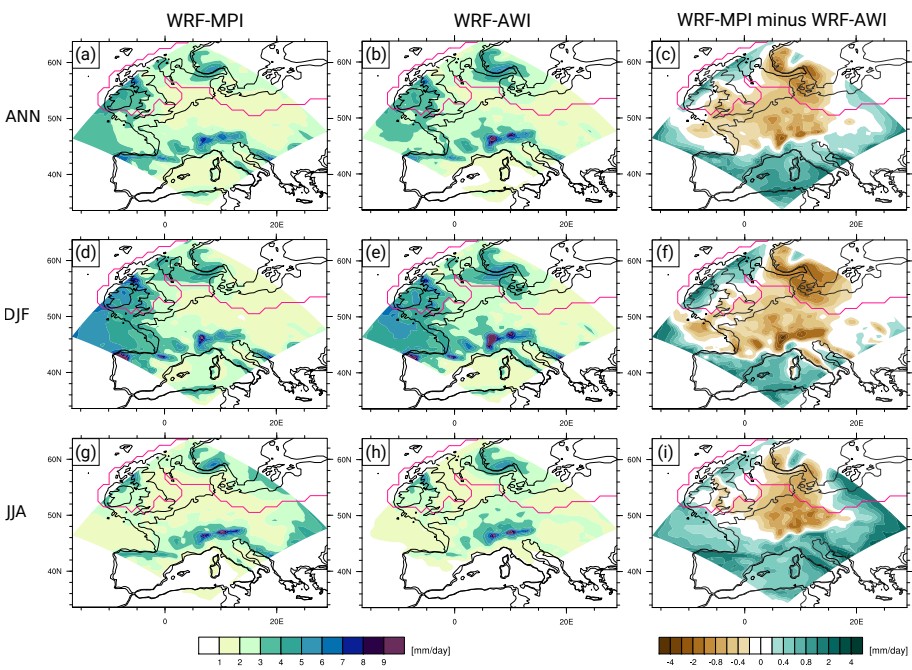

**Figure 4.** Distribution of total precipitation in annual *(a-c)* and seasonal winter *(d-f)* and summer *(g-i)* means as simulated with the regional WRF model with MPI forcing *(a, d and g)* and with AWI forcing *(b, e and h)* and their differences *(c, f and i)*. Black line: LGM coastline, pink line: LGM ice sheet.

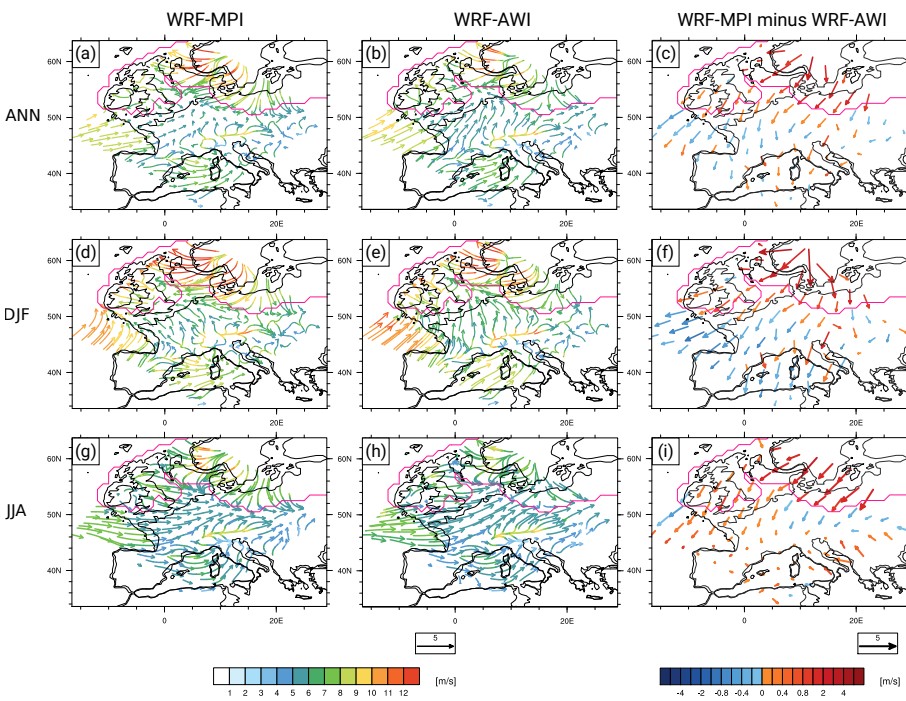

**Figure 5.** Distribution of 10 m wind speed in annual *(a-c)* and seasonal winter *(d-f)* and summer *(g-i)* means as simulated with the regional WRF model with MPI forcing *(a, d and g)* and with AWI forcing *(b, e and h)* and their differences *(c, f and i)*. Black line: LGM coastline, pink line: LGM ice sheet.

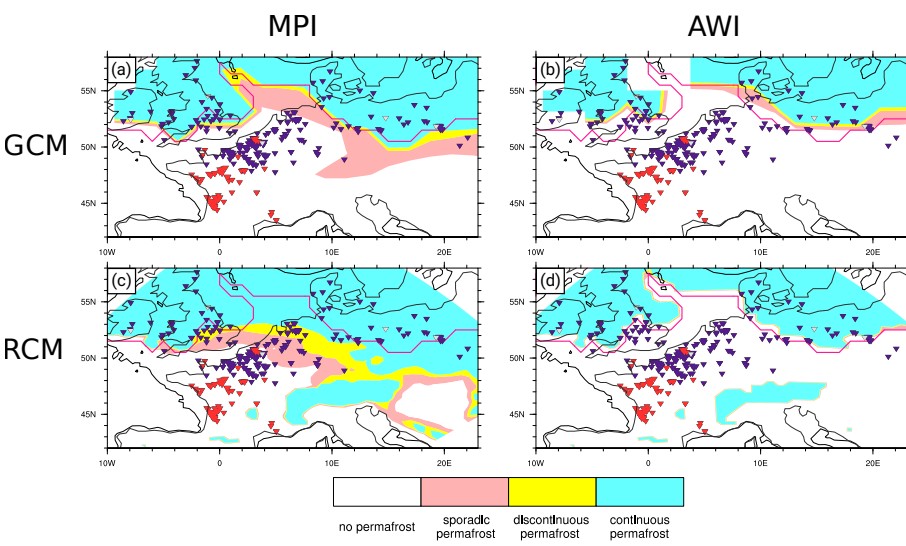

**Figure 6.** Permafrost distribution based on the surface frost index (SFI) in approximately 70 cm depth as simulated by the global MPI *(a)* and the global AWI simulation *(b)* and their respective regional counterpart *(c and d)*. Ice wedge pseudomorphs, composite, and sand wedges from Andrieux et al. (2016) and Isarin et al. (1998) are denoted by purple, grey, and red triangles respectively. Black line: LGM coastline, pink line: LGM ice sheet.

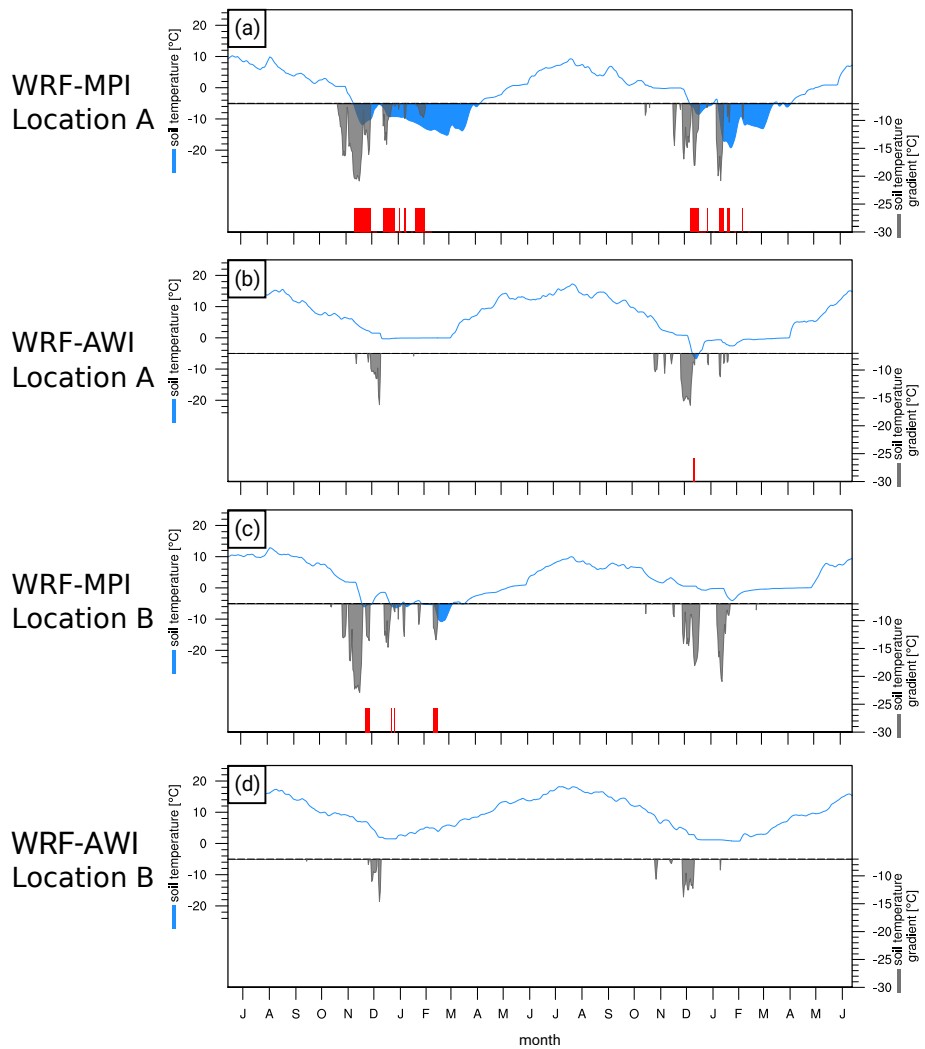

**Figure 7.** Time series of the daily mean soil temperature and soil temperature gradient in location A and B as denoted in Fig. 1 (A: model grid point 49.8° N, 2.49° E; B: model grid point 46.1° N, -0.33° E) as simulated in WRF-MPI *(a and c)* and in WRF-AWI *(b and d)* for two consecutive years. Blue lines show the development of the soil temperatures in layer 3 (70 cm). When the temperatures fall below -5 °C, the first condition for thermal contraction cracking after Matsuoka et al. (2018) is fulfilled, marked with blue shading and the reference line. The soil temperature gradient between the first layer (5 cm depth) and the third layer (grey shading) is only depicted when condition two after Matsuoka et al. (2018) is fulfilled, with a gradient below -7 °C m$^{-1}$. Red lines indicate the coincidence of the two conditions.

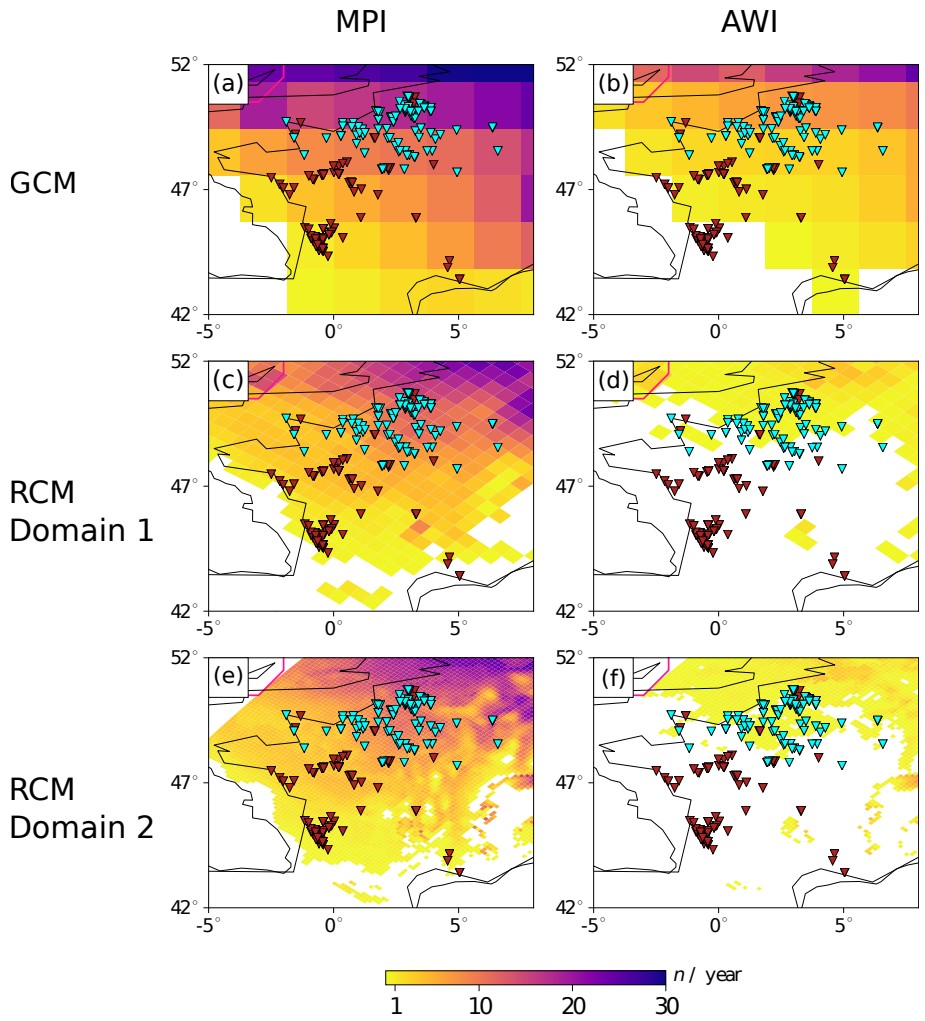

**Figure 8.** Heat maps of the mean number of days per year when the minimum conditions for thermal contraction cracking after Matsuoka et al. (2018) are fulfilled for each grid box in the global MPI *(a)* and AWI simulations *(b)*, and for the first domain of the regional WRF-MPI *(c)* and WRF-AWI *(d)* simulations, as well as for the second domain in WRF-MPI *(e)* and in WRF-AWI *(f)*. Ice wedge pseudomorphs and sand wedges from Andrieux et al. (2016) are highlighted with cyan and red triangles, respectively, only when located in France. Black line: LGM coastline, gray line: LGM ice sheet.