# Peer review of "A new perspective of permafrost boundaries in France during the Last Glacial Maximum"

_Climate of the Past, 2021_

## Author Comment (AC1)

**Final Response to Anonymous Referee #1**

**General comments:**

The general idea of the paper, i.e. testing climate model outputs for the LGM with the spatial distribution of permafrost features is very good. Not being a climate modeller, I cannot criticize very much that aspect of the paper. However, I suggest that the paper could be significantly improved by providing better information from the literature and the co-author's knowledge about the current climate-driven mechanisms that drive active processes, particularly frost cracking, ice wedge activity, growth and decay and cracking in non permafrost regions. A more thorough and clear presentation of the current driving factors and resulting features would help the reader understand the results of the research in the modelling context. This would also benefit the discussion section.

The authors could also justify some of their methodological choices. E.g. why those specific values for frost cracking (-5 °C at 1 m and -7 °C/m gradient) and not others? Why extract SFIs at 0,7 m and not in the air?

Several statements need to be clarified. See detailed comments below.

We thank the reviewer for his/her time invested to read the manuscript in such a careful and thorough manner. The suggestions and corrections led to further improvements in the manuscript. The comments have been carefully considered and responded. Please find below our response to each comment.

**Detailed comments:**

**Abstract:**

I suggest that the abstract should state more clearly what the objectives of the research are. They could (?) be stated as: 1- to evaluate the potential of regional climate model simulations to reconstruct the permafrost distribution in western Europe during the LGM, 2-apply to modelled data the experimentally known parameters for frost cracking to LGM in Europe. 3-identify the regional climate model of the LGM that best reproduces permafrost distribution and soil thermal contraction conditions as represented by fossil ice wedge casts and sand wedges.

We thank the reviewer for this comment. As suggested, we pointed out the objectives of our study more precisely. However, we do not fully agree with proposed objectives. Especially the third proposed objective would be interesting but is beyond the scope of our study, since we use the output of two different global climate model, but only one regional climate model.

Our second reviewer Jef Vandenberghe found "the objectives clearly described in the Introduction" (see his first general comment), therefore, we decided to keep the objectives as before and changed the second sentence of the abstract to

"In this work, we **aim to explore the possible benefit of using regional climate model data to improve the permafrost representation in France, to decipher how the atmospheric circulation**

**affect the permafrost boundaries in the models and to test the role of ground thermal contraction cracking in wedge development during the LGM.**"

Lines 8-9 : "Whereas" …the meaning of this sentence is confusing. Do you mean that the global model for the LGM does not fit with permafrost distribution and ground cracking conditions worldwide but that a regional model does it for Europe? I do not understand well here. Say in a better way.

The permafrost distributions that are derived from global climate model simulations under present day and paleo conditions often deviate from the actual permafrost occurrence or proxy based reconstructed distributions in Europe, but also around the globe at relevant locations. There are several publications with this focus (e.g., Kitover et al., 2013, Koven et al., 2012, Levavasseur et al., 2011, Saito et al., 2013, and Ludwig et al., 2017). This is also the case for the permafrost distribution and ground cracking regions in Europe derived from the global climate model simulations used in our study. However, we find improvement for the permafrost distribution and ground cracking regions in Europe derived from the regional climate model simulations over derivations from global climate model data.

In order to clarify this, we deleted this sentence and changed to subsequent text to:

"Given the appropriate forcing, an added value of the regional climate model simulation can be achieved **in representing permafrost and ground thermal contraction cracking**."

Line 11: Thermal contraction cracking occurring south of permafrost zone: This does not come as a surprise. Even nowadays, we get ground temperature conditions for frost cracking at places south of the permafrost region. But how deep is a question. It depends on the maximum depth reached by the freezing front locally.

Quite unexpected was the width of the latitudinal band south of the permafrost area that underwent thermal contraction cracking in France during the LGM. The simulations do not provide insight into the depth of ground cracking, but field data have shown that the depth of the frost wedges diminishes southwards to reach approximately 1.8 m at 45° N (Andrieux et al. 2016), which provides a minimal value for the depth reached by the freezing front (which was probably much deeper as showed by Wolfe et al. (2018).

We modified the sentence in the manuscript as follows:

"Furthermore, the model data provide evidence that thermal contraction cracking occurred in Europe during the LGM **in a wide latitudinal band** south of the probable permafrost border, in agreement with field data analysis."

**Introduction:**

Lines 17-20; current thawing of permafrost and carbon feedback. This is not pertinent for this paper. I suggest delete those lines.

With this paragraph we want to underline the general importance of the topic.

Line 23; replace "under" by "against which" they are well tested

The models get present-day conditions as input/forcing (e.g., greenhouse gas concentrations, distributions of vegetation and land use, solar constant), we therefore decided to keep the sentence without changes.

Line 38; add "a " colder, drier… period

We added "a" as suggested.

Line 59-60: provide a better explanation for ice wedge growth and origin of ice wedge casts: annual frost cracks that reach downward into the permafrost are a few mm wide. They get filled with snowmelt water that freezes into ice veins. Repeated cracking over years at the same location add ice veins that constitute ice wedges. Wedge casts (pseudomorphs) observed from the LGM in Europe were formed when the ice wedges melted and the cavities were filled by collapsing soil materials. I suggest you cite here Harry and Godsik (1988).

To better explain the development of ice wedges and ice-wedge pseudomorphs, we changed ll. 58-61 as follows:

"Annual frost cracks that reach downward into the permafrost are a few mm wide. They get filled with snowmelt water that freezes into ice veins. Repeated cracking over years at the same location add ice veins that constitute ice wedges (e.g., Harry and Gozdzik, 1988; Murton, 2013). Ice wedge pseudomorphs observed from the LGM in Europe were formed when the ice melted and the cavities were filled by collapsing soil materials."

Line 67-70: weird sentences here: limiting factor? This is complex language. Thermal contraction cracking is the causal factor that leads to ice wedge growth (see above); it is when the conditions for thermal contraction are not met that there is no cracking ... Ecological factors such as type of vegetation cover and thick snow cover often limit thermal contraction cracking when they prevent the cooling of the ground. Note that frost cracking also occurs widely in cold environments in roads and airport runways.

We agree with the reviewer, that thermal contraction cracking together with the occurrence of permafrost is both causal to ice wedge growth. In current Arctic regions, it is rather the lack of conditions that enable ground cracking that prevents ice wedge development, even in regions where permafrost is present. This is why we called thermal contraction cracking a limiting factor.

Our results show, however, that in Europe during the LGM, it is the other way around: According to our simulations, thermal contraction cracking was possible even south of the probable permafrost border. It was the lack of permafrost that prevented ice wedge growth.

In order to clarify this, we changed ll. 64-76 as follows:

"Active sand wedges are found today **primarily** in areas characterized by continuous permafrost and limited snow and vegetation cover **(i.e. the polar deserts),** and with local sources of aeolian sediments such as in Antarctica (Bockheim et al., 2009; Levy et al., 2008; **Murton et al., 2000;** Pewe, 1959). **Ground cracking is often restricted to the active layer (i.e. the surface layer subjected to seasonal freezing and thawing) in the areas underlain by "warm" permafrost (i.e. at a temperature close to 0°C) and south of the permafrost border. Thin cracks develop and are referred to as seasonal frost cracks.** However, Wolfe et al. (2018) showed that **large shallow** sand wedges can **also** develop in Canada in areas with deep seasonal ground freezing (i.e. without perennially frozen ground) in mineral soils close to dune fields**, which provide abundant sand to fill the cracks**.

Thermal contraction cracking of the ground **is the causal factor that leads to ice (or sand) wedge growth. Ecological factors such as type of vegetation cover and thick snow cover often limit thermal contraction cracking when they prevent the cooling of the ground. This is the case in current densely vegetated areas that insulate the ground and trap snow (e.g. shrub tundra and taiga; Kokelj et al., 2014; Mackay and Burn, 2002). Conversely, cracking can occur at low frequency in mid-latitude, cool temperate regions in grounds devoid of tall vegetation and snow, particularly in roads and airport runways (Barosh, 2000; Okkonen et al., 2020; Washburn, 1963)."**

Line 81: considered

We changed "considering" to "considered".

Lines 80-96: This paragraph is confusing. All approaches to map permafrost temperatures ( ex. 1D gridded models, TTOP and others require that a thermal offset factor (ex. n-factors) be applied to air temperatures (freezing and thawing degree-days) to map soil surface temperatures. SFI also. At line 93, without any context, I do not understand what is the deepest ground layer and why 5,7 m deep? Deepest relative to what? the deepest depth applied in Stendel & Christensen's model? Temperature in permafrost at that depth may be good to monitor or predict changes, but it tells little in terms of permafrost type or distribution contrary for instance to depth of 0 ºC thermal amplitude.

The deepest soil layer refers to the climate model, Stendel and Christensen (2002) used in their study.

Although the SFI was originally based on 2 m air temperatures, it is also applied to simulated ground temperatures by climate models (see for example Stendel and Christensen, 2002 and Ludwig et al., 2017). Climate models directly take into account snow cover and vegetation. Thus, additional thermal offset factors are not longer necessary.

In the description of the adaptions by Stendel and Christensen (2002), the deepest soil layer refers to the climate model, the authors used in their study.

For clarification, we changed the paragraph (ll. 101-105) as follows: "Slater and Lawrence (2013) weighted the snow depth for each month to consider snow accumulation effects, while Stendel and Christensen (2002) replaced the surface air temperature with the temperature of **their deepest simulated** ground layer (5.7 m deep) to investigate permafrost degradation due to current global warming. **They pointed out the advantage of taking simulated ground temperatures, where insulation effects of snow and vegetation cover are explicitly taken into account by the models, render empirical approaches redundant.**"

Lines 100-103: I suggest to rewrite the objectives as suggested as above in the abstract section. For instances objective 3 should be reformulated: We already know that thermal contraction is the process that drives cracking and ice wedge development. Should not the objective be to test how the spatial distribution of conditions for thermal contraction cracking modelled with a regional model of the LGM fits with observed distribution of relict wedge casts and relict frost cracks ???

In line with the second reviewer Jef Vandenberghe, we decided to keep the objectives without changes. Regarding the third objective, it is the focus on the LGM that is important. (See also answers to comments to the abstract.)

**Data and methods:**

Lines 110-139: I am not competent to criticize the choice of global and regional models.

The applied models are well established and widely used in both global and regional climate modeling communities. Thus, we think that the choice of the models is well justified and we do not expect incorrect statements due to insufficient/erroneous models.

Lines 140-143: The definition of permafrost zones according to MAATs at 2 m provides a good estimate of permafrost distributions. The SFI maybe does a somewhat better job to meet your objectives.

MAAT may be a good and simple approach for the present day permafrost distribution, however, our results show that the permafrost boundaries by MAAT under LGM conditions do not seem reasonable as they agree more or less with the ice sheet margin.

Line 149: SFI calculated from outputs at 70 and 78 cm are not the same as the original SFI concept calculated from air surface temperature. The selection of this depth in this paper needs to be justified. Why not nearer to the surface?

However TTOP is also based on DDF and DDT and of easy use over a gridded domain. You could apply some general soil data to infer soil thermal conductivities. The maps of permafrost distribution and temperature in the LGM in Europe would likely be better. See Way and Lewkowicz 2016 in Canadian Journal of Earth Sciences for an actual modelled application over Quebec-Labrador. (Maybe for another paper?)

As mentioned in our answer to your comment to ll. 80-96, we apply ground temperatures to the SFI, so that additional thermal offset factors are not necessary (but directly taken into account by the models). We use these depths because global and regional model layers are closest to each other and are thus better comparable. We clarified the advantages in the introduction.

We tested the TTOP method as suggested, an example can be found below (Fig. 1). The resulting permafrost distributions resemble those derived with MAAT and do not provide additional information. Furthermore, it is not necessary to infer soil thermal conductivities, because they are explicitly considered in the model, dependent on the respective soil type that is given to the model for each grid point. We therefore decided against including this method to the paper, but to explain our land surface model in more detail, to include further references in the Data and Methods section (ll. 145-147) and we added a figure with snow height distributions of the regional simulations to the Supplementary.

[Figure]

Figure 1: (a) Distribution of TTOP based on the WRF-MPI simulation and (b) the resulting permafrost distribution with TTOP = -6 °C.

Lines 151-154:

Why do you retain only those values of -5 °C at 1 m and -7 °C/m as thermal gradient? They occur only for one cracking event at one of Matsuoka et al's three measurement sites in Svalbard. They have another site with values as modest as -2,8 °C and 1.1 °C/m. Their general (averaged) values for frost cracking events to occur are -20 °C at the ground surface, -10 °C at permafrost top (or 1 m deep) and a gradient Ë -10 °C/m.

In their paper, Matsuoka et al., (2018) identified values of T100 < -5 °C and GAL < -7 °C/m as minimum conditions for thermal contraction cracking at one of their three measurement sites. The very

high values during one cracking event at this site (T100 = -2.8 °C and GAL = 1.1 °C/m) were exceptional. The authors constrain that those minimum conditions might represent shallow cracking within the active layer, which we consider being adequate to analyse the sand wedges south of the permafrost border in southwestern France.

The slightly lower values (T100 = -10 °C and GAL = -10 °C/m) are conditions for intensive, deep cracking reaching the ice wedge. Heat maps with these conditions as shown below (Fig. 2) are now also included to the Supplementary and are compared against the ice-wedge pseudomorph distribution in France.

We agree that this decision needs further clarification and included the following explanation in the Data and Methods section (ll. 168-171):

"These minimum values might represent shallow cracking within the active layer/seasonally frozen layer and can be compared against the sand wedge distribution. Conditions for intensive and deep thermal contraction cracking (T100 = -10 °C and GAL = -10 °C/m) are tested in regard to the ice-wedge pseudomorph distribution in France."

Did you make any calculation (interpolations) to adjust the selected 78 cm depth output of your climate models to the general or the minimal values of Matsuoka et al.?

The empirically determined conditions for thermal contraction cracking allow us estimations on where ground cracking could occur in Europe during the LGM. We are aware of the differences between ground characteristics of the measurement sites (in Svalbard, Norway) and our region of interest.

We think that an interpolation would rather lead to an impression of more precision than we could provide with our estimations. Interpolating would also lower the comparability between the global and the regional simulations: We would have to interpolate between data from 70 cm and 150 cm depth of the regional model, but between data from 78 cm and 268 cm of the global model.

It should also be explained in the paper that these values apply to some single frost cracking events over periods of about 3 days in a given winter.

Matsuoka et al. (2018) used different parameters to determine conditions for thermal contraction cracking: temperature in 1 m depth and thermal gradient in the active layer on the one hand, and a cooling rate at the surface on the other hand. The 3-day period only applies to the cooling rate at the surface.

Those values should lead to frost cracking in soils both in permafrost regions (often) and in non-permafrost regions (occasionnally). Pseudomorphs were ice wedges in permafrost. Small frost cracks may have occurred in seasonally frozen ground. A clear explanation in the paper would better support your interpretations.

With the additional explanation and the distinction between conditions for shallow and for deep thermal contraction cracking in the Data and Methods section as introduced above, we hope to have clarified this point.

[Figure]

*Figure 2: Heat maps of the mean number of days per year when the conditions for deep thermal contraction cracking after Matsuoka et al. (2018) are fulfilled for each grid box in the global MPI (a) and AWI simulations (b), and for the first domain of the regional WRF-MPI (c) and WRF-AWI (d)simulations, as well as for the second domain in WRF-MPI (e) and in WRF-AWI (f). Ice wedge pseudomorphs and sand wedges from Andrieux et al. (2016) are highlighted with cyan and red triangles, respectively, only when located in France. Black line: LGM coastline, gray line: LGM ice sheet.*

**Results:**

We understand the WRF-MPI model fits better with field observations of pseudomorphs and fossil cracks. An interesting result.

Thanks. This again shows the importance of using more than one model to simulate the potential permafrost boundaries.

Line 210: the 70 cm depth is representative of what? I rather understand it is the depth you selected in the model output to check against SFIs that are surface values. I suggest you write "selected" or "depth chosen as representative."

The ground in our regional climate model consists of four ground layers. Each ground layer considers a range of depths and is then representative for a specific depth (5 cm, 25 cm, 70 cm, and 150 cm). This is described in the Data and Methods section (see ll. 151-152).

We changed the sentence to "In eastern Europe, the distribution of ice wedge pseudomorphs (Isarin et al., 1998) strictly overlaps that of modelled continuous permafrost in the **selected** layer with a depth of 70 cm."

Line 243: "The SFI is suitable to infer LGM permafrost from model data". With your simulated climate data, you might have had better results in representing permafrost distribution with the TTOP model (based also on freezing and thawing degree-days). This would have allowed you to map temperatures at the top of permafrost over the regional domain and compare it with air temperatures at 2 m above ground, + calculate the surface offset. By selecting a value of about -6 ºC for TTOP, you might be close to the southern limit of ice wedges active during the LGM

See answer to comment to l. 149.

Lines 255: again, here I do not understand your concept of "limiting factor". Thermal contraction cracking is the CAUSAL factor for developing ice wedges in cold enough permafrost AND shallower thinner sand wedges above warm permafrost and in the seasonal frost zone. Simply put, thermal conditions for frost cracking were present in the LGM. I suggest just avoid too much language complexity.

The concept of permafrost occurrence or thermal contraction cracking as limiting factor for ice wedge growth is explained in the answer to the comment to ll. 67-70.

To account for this comment and for the comments to ll. 256-258 and l. 256, we rewrote ll. 283-291 as follows:

"Contrary to what occurs today in large Arctic areas underlain by permafrost, where ground insulation by dense vegetation (shrub tundra, taiga) and snow cover prevents ground cracking and limits the growth of ice wedges (existing ice wedges that have formed in relation to different climatic or ecological conditions do not melt but are dormant), ice wedge growth in permafrost areas in France during the LGM was rapid because thermal conditions leading to ground cracking occurred with high

frequency. Large ice wedges (which after thawing developed into recognizable pseudomorphs) would have formed in permafrost where it was cold enough in winter to crack. Simulations show that periods of winter ground temperatures below -10 °C at 1 m depth could occur in the discontinuous and sporadic permafrost zone, suggesting that thermal contraction cracks were possibly not restricted to the active layer but could propagate into the permafrost in these areas leading to the development of ice wedges."

Line 256-258: " ice wedges would have developed in the discontinuous and sporadic permafrost zone, with the limiting factor being only the ability for ice to be preserved from year to year". This sentence lacks logics. If ice veins melt one year or another then forget about development of ice wedges. However, in limiting conditions, ice wedges may crack only every other year or become dormant in warm permafrost for periods without melting.

For ice-wedges to form we need cold enough permafrost temperatures (around -10 °C) in winter so that the cracks propagate to depths of several meters. The coalescent veins (the wedge) do not melt BECAUSE they are IN the permafrost.

With the sentence above, we meant that ice wedges grew in the permafrost, including in areas where it was discontinuous or sporadic.

In agreement with the simulations, periods of winter ground temperatures below -5 °C at 1 m depth could occur in the discontinuous and sporadic permafrost zone, suggesting that thermal contraction cracks could propagate into the permafrost where it existed, leading to the development of sand and ice wedges.

Line 256: why "densely vegetated Arctic areas"?? ice wedges are found under various tundra types (polygonal, tussocks, moss, lichens, patterns of lichens and shrubs, etc.)

Large Arctic areas with continuous permafrost under shrub tundra and taiga are today characterized by dormant ice wedges as ground thermal contraction occurs only very infrequently. The LGM pattern in Europe was quite different.

**Figures:**

Figure 6: SFI applied at the 70 cm depth . In the original concept, SFI is based on degree-days in air temperatures (2 m above ground.) Then why not present a permafrost map based on SFI with air temperatures.

See answer to comment to ll. 80-96

Similarly why not show a permafrost map based on MAATs for comparison (as in lines 140-143).

We show the permafrost distribution based on MAAT in the Supplementary (Fig. S4) and describe it in the text (see ll. 239-244). Since permafrost based on MAAT is limited to the ice sheet, we did not include the figures in the manuscript.

Also, it seems to me that the size and shapes of the sand wedges from Andrieux should be mentioned in the text. Large sand wedges could have developed in dry very cold environment (polar desert conditions) but thin frost cracks could have open in discontinuous permafrost or even in seasonally frozen ground.

The size and shape of the sand wedges are detailed in Andrieux et al. (2016) and it is beyond the scope of the present paper to discuss this topic. Large sand wedges can occur at low latitude, i.e. sand wedges up to 1 m wide have been found in southwest France near 45° N in the periphery of coversands. Optically Stimulated Luminescence dating of the sand fill by Andrieux et al. (2018) have demonstrated that these large epigenetic sand wedges resulted from repeated periods of growth throughout the Last Glacial. Therefore, width is not a good criterion for assessing climate at the time of wedge formation.

I also think that reasons why the more or less good fit between spatial distribution of fossil features and (even) the best option of permafrost map should be discussed in the paper.

In the Summary and Discussion section, we discuss possible reasons for differences between proxy and model data. (See ll. 312-320; beginning with "Various factors may account for a remaining gap between proxy and model data. These factors include:")

Figure 7: values (months?) are needed on the time scale.

We changed the time scale as suggested.

---

## Author Comment (AC2)

**Final Response to Jef Vandenberghe**

We thank Jef Vandenberghe for his time invested to read the manuscript in such a careful and thorough manner. The suggestions/corrections led to further improvements in the manuscript. The comments have been carefully considered and responded. Please find below our response to each comment.

**General comments:**

1. I find the objectives clearly described in the Introduction and adequately discussed in the final Discussion. The methodology is well explained also for non-specialists in modelling (section 2) The analysis of the results is fine to me, although they may be better structured and organized.

The manuscript was revised carefully based on your review and on that of one anonymous reviewer. We hope that these improvements also helped the structure of our analysis.

2. Also the first part of the Abstract (l 1-8) suffers from poor cohesion.

We changed the first part of the abstract and added the aims of our study explicitly.

3. Since I am not a climate modeller nor permafrost modeller, I have no real comments on those aspects.

We chose both global climate models and the regional climate model that are well established and widely used. Thus, we think that the choice of the models is well justified and we do not expect incorrect statements due to insufficient/erroneous models.

4. Section 3.1: state why the analysis in this section is relevant for the objective of this study.

We included the following paragraph to point out the relevance of this section:

"In this section, we present the large-scale characteristics of the LGM climate derived from global climate model data that are used for dynamical downscaling in comparison with the respective PI simulations. It is important to investigate the climatic mean state and possible biases of the global projections in order to be able to interpret the regional simulations accurately."

5. wind circulation:

-l 50-52: please allude here to the hypothesis of proxy evidence for northern and western winds during LGM in NW Europe (eolian sands and loess and morphology) as forwarded in papers by Renssen et al 2007 (JQS22 (3), p 281-293) and papers by Schwan (Sedimentary Geology).

We added the suggested literature and the hypothesis on westerly to northwesterly winds in ll. 51-52.

-Further in l 193-203 you derive and discuss especially western and d northwestern wind, This seems contradictory to me. Please, explain better.

Winds southward of the Fennoscandian Ice sheet are easterlies/northeasterlies. These are the strongest in the model domain in annual mean and each season for one regional simulation (WRF-MPI) and in winter for both simulations. We included this statement in ll. 216-217. Additionally, we corrected the sentence in l. 219 to "whereas in WRF-MPI the winds have a more **southwestern** component".

**Minor comments:**

-l 6 and 7: 'large-scale circulation' and 'LGM climate': obtained by modelling?

Yes, climate and large-scale circulation are obtained by the climate models. Therefore, we changed the sentence to

"Our results show that the permafrost **extent and ground cracking regions deviate from proxy evidence when the simulated large-scale circulation in both global and regional climate models favours** prevailing westerly winds."

-l 12-13: sentence is vague

We changed the sentence to "This enables the reconsideration of **the role of sand wedge casts to identify past permafrost regions**."

-l 17: this definition of permafrost is much older than 2005

This reference is a glossary on permafrost and related terms which was revised in year 2005, but was originally based on a publication from 1988. We included "e.g." in front of the citation.

-l 31: '130 ka': at the coldest phase of LGM

We changed the sentence to

"**During the coldest phase of the LGM,** the sea level was about 130 m lower than today..."

-l 38: these climate comparisons concern France

In ll. 29-40, we refer mainly to global climate changes.

-l 56: suggest to refer to refer also to Huijzer & Isarin 1997 in QSR and Vandenberghe 1983 in Polarforschung

We included the references as suggested.

-l 55-57: 'ice-wedge pseudomorphs most reliable for pf reconstruction'. Ok, but why not mentioning the large cryoturbations (as evidence of former permafrost degradation) and the deep large sandwedges that formed in permanently frozen subsoil (e.g. papers by Ghyssels in PPP: LGM wedges in Belgium) and by Murton (Canada).

We mention that there is "a variety of fossil periglacial features", where cryoturbations are also included and that among those features, "ice wedge pseudomorphs are the most reliable". As the later analysis focuses on ice-wedge pseudomorphs and sand wedges, we decided to point out their role on the derivation of past permafrost distributions, but the references we cite at this point, also address cryoturbations (e.g., Bertran et al., 2014, Vandenberghe, 1983, and Vandenberghe et al., 2014).

-l 65-66: (see also comment in l 254) these wedges are formed in the subsoil that is affected by deep winter frost (and thus are shallow) as is also stated in l 254-255.

We included "shallow" to clarify that we are aware of this fact.

-l 73: please insert 'as reported by' after 'earliest reconstructions'

Changed as suggested.

-l 89 and 243: 'effects of snow for SFI': OK, although estimates of snow presence/thickness at LGM are speculative. But, if you mean the use of SFI only in models, please state that clearly.

Indeed, we only apply the SFI on climate model data. We stated this more clearly in l. 97 and in the Data and Methods section.

-l 143: I suppose you jump here to modelling experiments as SFI is difficult to estimate from proxy data in paleo-records. Thus, I suggest starting a new paragraph here.

Also for MAAT, we use climate model data. We clarified this in l. 155.

-l 254: I suggest also to refer to similar structures on the Ordos Plateau in China (small-sized or shallow sand wedges formed under conditions of deep seasonal frost): Vandenberghe et al 2004 in PPP and 2019 in QSR.

The suggested reference is introduced in ll. 85-88 as follows "A similar pattern has also been highlighted in China by Vandenberghe et al. (2019). The sand wedges reach up to 1m wide in southwest France near 45°N in the periphery of coversands. Optically Stimulated Luminescence dating of the sand fill by Andrieux et al. (2018) have demonstrated that these large epigenetic sand wedges resulted from repeated periods of growth throughout the Last Glacial."